# Predicting Immunogenic Epitopes Variation of Envelope 2 Gene Among Chikungunya Virus Clonal Lineages by an In Silico Approach

**DOI:** 10.3390/v16111689

**Published:** 2024-10-29

**Authors:** Sung-Yeon Cho, Dong-Gun Lee, Jung Yeon Park, Won-Bok Kim, Raeseok Lee, Dukhee Nho, Eun-Jee Oh, Hyeyoung Lee, Chulmin Park

**Affiliations:** 1Vaccine Bio Research Institute, College of Medicine, The Catholic University of Korea, 222 Banpo-daero, Seocho-gu, Seoul 06591, Republic of Korea; cho.sy@catholic.ac.kr (S.-Y.C.); symonlee@catholic.ac.kr (D.-G.L.); tramppark@gmail.com (J.Y.P.); skaks301@naver.com (W.-B.K.); misozium03@catholic.ac.kr (R.L.); nhodh@catholic.ac.kr (D.N.); 2Division of Infectious Diseases, Department of Internal Medicine, College of Medicine, The Catholic University of Korea, Seoul 06591, Republic of Korea; 3Department of Laboratory Medicine, College of Medicine, The Catholic University of Korea, Seoul 06591, Republic of Korea; ejoh@catholic.ac.kr; 4Department of Laboratory Medicine, International St. Mary’s Hospital, College of Medicine, Catholic Kwandong University, Incheon 22711, Republic of Korea; shomermaid@gmail.com

**Keywords:** chikungunya fever, Chikungunya virus, Alphavirus, envelope 2, in silico approach, MHC binding epitopes, molecular docking

## Abstract

Chikungunya virus (CHIKV), responsible for a mosquito-borne viral illness, has rapidly spread worldwide, posing a significant global health threat. In this study, we explored the immunogenic variability of CHIKV envelope 2 (E2), a pivotal component in the anti-CHIKV immune response, using an in silico approach. After extracting the representative sequence types of the CHIKV E2 antigen, we predicted the structure-based B-cell epitopes and MHC I and II binding T-cell epitopes. Variations in key T-cell epitopes were further analyzed using molecular docking simulations. We extracted 258 E2 gene sequences from a pool of 1660 blast hits, displaying homology levels ranging from 93.6% to 100%. This revealed 44 sequence types, each representing a unique genetic variant. Phylogenetic analysis revealed distinct geographically distributed clonal lineages (clades I-IV). The B-cell linear and discontinuous epitopes demonstrated a similar distribution across the E2 protein of different strains, spanning domains A, B, and C, with some slight variations. Moreover, T-cell epitope prediction revealed eight conserved MHC class I hot spots and three MHC II hot spots, displaying variations among lineages. Among clade II strains, there were significant variations (N5H, S118G, G194S, L248F/S, and I255V/T) observed in epitopes, distinct from strains belonging to other lineages. Additionally, molecular docking showed that variations in MHC I epitopes across clonal lineages induced changes in the structure of the peptide–MHC complexes, potentially resulting in immunogenic disparities. We expect that this in silico approach will serve as a complementary tool to experimental platforms for exploring immunogenic variation or developing biomarkers for vaccine design and other related studies.

## 1. Introduction

Chikungunya virus (CHIKV) possesses a single-stranded positive-sense RNA genome of approximately 12 kilobases, and is classified within *Togaviridae Alphavirus*. CHIKV is primarily transmitted to humans by *Aedes* mosquitoes, particularly *Aedes aegypti* and *Aedes albopictus*, which predominantly inhabit tropical and subtropical regions [1,2]. CHIKV infection typically manifests as a febrile illness known as chikungunya fever, which is characterized by symptoms such as headache, myalgia, rashes, and severe arthralgia [3,4,5]. Phylogenetic analysis of CHIKV revealed four primary genotypes: West African (WA), East Central South African (ECSA), Indian Ocean and Indian (IOI), and Asian genotypes [6,7,8,9]. Recently, CHIKV has disseminated into the Western Hemisphere (Europe and North America) through viraemic travelers [10]. CHIKV shares the same vectors with Flaviviruses, and factors such as global warming and an increasing number of viraemic travelers may expedite its global dissemination. The CHIKV genome encodes non-structural (nsP1-4) and structural proteins, which include the capsid and four envelope proteins comprising two glycoproteins (E1 and E2) and two small proteins (E3 and 6K) [11].

The main immunogenic proteins include E2, E3, capsid, and nsP3, with the E2 glycoprotein playing a pivotal role in the anti-CHIKV response throughout disease progression [12]. Because of its high immunogenicity, E2 is considered a crucial antigen for enhancing diagnostic techniques and developing a subunit vaccine [13,14].

The US FDA recently approved the first CHIKV vaccine, Ixchiq, previously known as VLA-1553. VLA-1553 is derived from the La Reunion strain (LR2006-OPY1) of the ESCA lineage and is a live-attenuated vaccine produced through deletion of nsP3 [15]. However, the diverse geographical lineages of CHIKV encompass multiple genotypes, leading to uncertainty regarding potentially immunogenic variants. Therefore, we aimed to explore the relationship between genetic variation and immunogenic diversity within clonal lineages.

In this study, we delineated an in silico approach for processing and analyzing extensive genetic variations and immunogenic diversity. Our aim was to improve the design and selection of immune biomarkers that are effective against clinically relevant strains.

## 2. Materials and Methods

### 2.1. Phylogenetic Analysis of CHIKV E2 Genes from Global Databases and Production of Representative E2 Recombinant Proteins

A total of 258 representative CHIKV E2 amino acid sequences (each comprising 423 amino acids) were curated from a pool of 1660 sequences identified using the position-specific iterated blastp (psi-blastp) algorithm. The selection criteria involved thresholds for e-value and bit score, with sequences containing gaps excluded from consideration. Additionally, the geographic origin of each sequence was determined. The sequence types (STs) were determined according to their homology and the unique amino acid sequence. Using Clustal and the Tamura-Nei model [16], alignment was performed using megalign pro (Dnastar Inc., Madison, WI, USA) or the MEGA7 software [17]. By the Maximum Likelihood method [18], we performed phylogenetic analysis using the MEGA7 software.

The synthesized E2 sequences from three representative strains (ESCA, IOI, and Asian strains) were cloned into the pET30a vector and subsequently transformed into the *Escherichia coli* Lemo21 (DE3) strain (New England BioLab Inc., Ipswich, MA, USA). Expression and solubility of the recombinant proteins were evaluated using SDS-PAGE. Each E2 recombinant protein was purified using an SDS lysis buffer, following a previously described method [19]. The purified E2 antigens were screened using anti-CHIKV monoclonal antibodies (mAbs) 6A11 and 11E7 (Kerafast Inc., Boston, MA, USA).

### 2.2. Predicting E2 Structure and Immunogenic Epitopes Using In Silico Approach

The trRosetta server (https://yanglab.qd.sdu.edu.cn/trRosetta/ [assessed on 24 April 2024]) [20] was used to predict the structure of each E2 protein from the ECSA (NP 690589), IOI (EF210157), and Asian strains (ACY66830). Each model was predicted based on the highest average probability of the top-ranked distances and the estimated TM-score. The confidence levels of the predicted models were evaluated using the estimated TM-score. Subsequently, models were constructed using restraints derived from both deep learning predictions and homologous templates. The predicted structures were compared with the mature envelope glycoprotein complex of CHIKV (PDB ID 3N42, https://doi.org/10.2210/pdb3N42/pdb) [21].

In addition, we investigated the immunogenic epitopes of each E2 using an immune epitope database and analysis resource (IEDB, http://www.iedb.org [assessed on 3 April 2024]). We predicted the MHC class I and II binding epitopes of E2 using the NetMHCPan 4 and NetMHCIIPan 4 [22] based on a reference panel of HLA alleles [23,24]. The reference panel of class I human leukocyte antigen (HLA) molecules included the following: HLA-A*01, HLA-A*02, HLA-A*03, HLA-A*11, HLA-A*23, HLA-A*24, HLA-A*26, HLA-A*30, HLA-A*31, HLA-A*32, HLA-A*33, HLA-A*68, HLA-B*07, HLA-B*08, HLA-B*15, HLA-B*35, HLA-B*40, HLA-B*44, HLA-B*51, HLA-B*53, HLA-B*57, and HLA-B*58.

The reference panel of class II HLA molecules included the following: HLA-DRB1*01, HLA-DRB1*03, HLA-DRB1*04, HLA-DRB1*07, HLA-DRB1*08, HLA-DRB1*09, HLA-DRB1*11, HLA-DRB1*12, HLA-DRB1*13, HLA-DRB1*15, HLA-DRB3*01, HLA-DRB3*02, HLA-DRB4*01, HLA-DRB5*01, HLA-DQA1*05/DQB1*02, HLA-DQA1*05/DQB1*03, HLA-DQA1*03/DQB1*03, HLA-DQA1*04/DQB1*04, HLA-DQA1*01/DQB1*05, HLA-DQA1*01/DQB1*06, HLA-DPA1*02/DPB1*01, HLA-DPA1*01/DPB1*02, HLA-DPA1*01/DPB1*04, HLA-DPA1*03/DPB1*04, HLA-DPA1*02/DPB1*05, and HLA-DPA1*02/DPB1*14; The predicted epitopes binding to MHC molecules were ranked based on their IC50 values (IC50 < 50 nM, high affinity; <500 nM, intermediate affinity; and <5000 nM, low affinity) and percentile ranks, which were generated by comparing the IC50 values of each peptide with those of random peptides from the SWISSPROT database. Epitopes with a percentile rank of <1 and an IC50 ≤ 50 nM were selected for further analysis.

B-cell linear and discontinuous epitopes (discotopes) based on the structural accessibility were predicted based on their structures by ElliPro and Discotope (IEDB, http://www.iedb.org [assessed on 30 August 2024]) [25,26]. The discotopes were predicted using the discotope score, which combines contact numbers (the number of Cα atoms in the antigen within a distance of 10 Å of the Cα atom of the residue) with propensity (reflected in amino acid epitope log-odds ratios). An epitope was considered positive if the score exceeded the threshold value of −3.7.

The predicted B- and T-cell epitopes were compared with experimental data from 249 T-cell epitope assays (e.g., IFN-gamma release ELISPOT, and biological activity assays) and 653 B-cell epitope assays (e.g., ELISA, microarray, and flow cytometry) for CHIKV structural and non-structural proteins, as available in the IEDB database.

### 2.3. Predicting Structure Docking

We assessed the interaction between E2 epitope peptides and HLA molecules using molecular docking predictions through Galaxypepdock by combining information on similar interactions found in the structure database and energy-based optimization (https://galaxy.seoklab.org/cgi-bin/submit.cgi?type=PEPDOCK [assessed on 17 April 2024]) [27]. The structures of MHC class I HLA-A2 (PDB ID 3MRB, https://doi.org/10.2210/pdb3MRB/pdb) and MHC class I HLA-A30 (PDB ID 6J1W, https://doi.org/10.2210/pdb6J1W/pdb) were obtained from the RSCB PDB server (https://www.rcsb.org/ [assessed on 3 March 2024]). Each simulation generated 10 models, and the models with the highest structure similarity (TM-score) and interaction similarity score (similarity between the HLA structure amino acid residues and the peptide amino acid) were selected. Additionally, the contact map depicting the interface between the receptor and the peptide was predicted by cluster density and root mean square deviation (RMSD) using CABS-dock online (https://biocomp.chem.uw.edu.pl/CABSdock/ [assessed on 28 April 2024]) [28]. The distance map was generated considering only Cα atoms located within a 5–15 Å range. Docking models were initially filtered from each of the 10 trajectories, and the selected models were clustered using the k-medoids procedure (k = 10).

Figure 1 presents an overview of the in silico process.

## 3. Results

### 3.1. Phylogenetic Analysis of CHIKV E2 and Immunogenicity Screening of E2 Recombinant Protein

Using psi-blastp retrieval, we acquired 258 sequences of CHIKV E2 genes from a pool of 1660 hit sequences, excluding those with gaps in the alignment. The selection criteria were based on percentage identity and bit score, resulting in a homology ranging from 100% to 93.6%.

Among these, 44 distinct STs with unique amino acid sequences were analyzed, and a final alignment was performed. The alignment and variation of the mature E2 sequences from 44 STs are shown in Appendix A. The genetic distances among these sequences ranged from 0.0000 to 0.0898. Phylogenetic analysis using the maximum likelihood method revealed variations in the geographical distribution of CHIKV (Figure 2). Within one of the major clusters, the E2 gene of IOI strains was closely related to that of certain Asian strains (Asian I cluster). In clade II, Asian strains (Asian II cluster) predominated, whereas Central South American, IOI, and Central strains were also observed (Figure 2). These findings highlight the genetic diversity of CHIKV E2 and the variations in its geographical distribution, which contribute to the genetic relationships and variations among CHIKV strains across different regions. Clades were distinguished based on a 0.02 sequence difference, leading to the definition of four distinct clades. An analysis of the homology within E2 of clades I and II, which contain a diverse range of genotypes (Figure 2), revealed that clade I exhibited sequence identity ranging from 99.57% to 100%, while clade II showed sequence identity ranging from 96.22% to 100%. The IOI strains were classified into clade I, Asian strains into clade II, ECSA strains into clade III, and West African strains into clade IV. Based on these clades and their frequencies, we selected three E2 candidates from three clades (ECSA strain: GenBank accession no. NP 690589, IOI strain: no. EF210157, and Asian strain: no. ACY66830).

The antigenicity of the purified recombinant E2 proteins was evaluated by Western blotting using the anti-CHIKV monoclonal antibodies 6A11 and 11E7 (Appendix A). All recombinant E2 antigens demonstrated specific reactivity with the 11E7 anti-CHIKV IgG monoclonal antibody (mAb), while only the Asian E2 antigen exhibited strong reactivity with the 6A11 anti-CHIKV IgG mAb. These results suggest potential antigenic variation among E2 antigens from different CHIKV clonal lineages.

### 3.2. Prediction of CHKV E2 Structure and B-Cell Epitope Using an In Silico Approach

The E2 structures of representative strains, including the ECSA, IOI, and Asian strains, were predicted (Figure 3A and Appendix A). The confidence of the predicted models was evaluated using estimated TM-scores, which were 0.849, 0.850, and 0.849, respectively. Comparison of each predicted structure with the CHIKV mature envelope glycoprotein complex revealed the presence of tdomain A (linked to E3 via the furin loop), domain B, domain C (associated with E1 in the mature complex), and the C-terminal region (Figure 3A). The three predicted structures (ECSA, IOI, and Asian strains) exhibit strikingly similar configurations (Figure 3A and Appendix A). The C-terminal region of E2 was found to be cleaved and subsequently joined to form the mature complex (E3-E2-E1) (Figure 3A).

We performed a predictive analysis of B-cell linear epitopes from the structure of the CHIKV E2 protein in representative strains using the IEDB (ElliPro) tool. The B-cell epitopes of E2 in each strain were found to be located in similar regions, although slight variations were observed in the profiles of the predicted epitopes among strains (Table 1). These epitopes were distributed across domains A, B, and C. Additionally, discotope analysis based on structural data revealed a similar distribution of epitopes across the E2 protein of different strains, including domains A, B, and C, and identified common amino acid residues shared with linear epitopes (Table 1 and Appendix A).

We compared the predicted B-cell epitopes with the immunological assay results for the CHIKV E2 protein obtained from the IEDB. A total of 90 linear epitopes and 41 discotopes were identified across 653 assays, corresponding to the mature E2 protein (Appendix A). The B-cell linear epitopes predicted in this study shared significant sequence similarity with those identified in the positive experimental data (Table 1 and Appendix A). Additionally, the positions of the predicted discontinuous epitopes (distopes) corresponded with those identified in the positive experimental data (Appendix A).

### 3.3. Prediction of T-Cell Epitope in CHKV E2 and Molecular Docking

T-cell epitopes for MHC class I and II binding were predicted from the mature E2 protein of each strain using IEDB. The predicted peptides were filtered based on a percentile rank below 1 (with smaller values indicating better binding activity) and an IC50 value of ≤50 nM for MHC binding. The prediction results for MHC class II binding peptides indicated that core peptides binding to MHC class II molecule are conserved in ESCA, IOI, and Asian strains (Table 2; 3–20 aa, 232–246 aa, and 280–299 aa, respectively), albeit with slight variations among strains. We identified two hot spots for MHC II binding, which were primarily located in the domains A and C. We identified three hot spots of MHC II binding, which were primarily located in the domains A and C (Figure 3B).

The prediction of MHC class I binding epitopes revealed that each epitope matching with the HLA-A and B reference set is 65 in the ESCA strain, 52 in the IOI strain, and 64 in the Asian strain (Appendix A). Additionally, eight hot spots of MHC I binding epitopes were identified across all domains of the mature E2 protein (Figure 3). When restricted to HLA-A*02, slight variations were observed among mature E2 proteins of the geographic lineage strains (Table 3).

However, mutation profiling at hot spots across 44 distinct STs (Appendix A) revealed that the majority of variations occurred in clade II STs, specifically, at positions T2, N5, S118, G194, L248, and I255 (Figure 3B), while few variations were detected in other STs.

Analysis of the predicted regions with a high distribution of MHC I and II hot spots and discotopes revealed that these regions predominantly overlapped in E2 domain A (MHC I and II hot spot 1–discotope [amino acid 1–13], MHC I hot spot 6/MHC II hot spot 2–discotope [230–256]) and domain C (MHC I hot spot 7 and 8/MHC II hot spot 3–discotope [267–318]) (Figure 3B).

We compared the predicted T-cell epitopes with the immunological assay results for the CHIKV E2 protein obtained from the IEDB. A total of 36 epitopes corresponding to the mature E2 protein were identified in 249 assays for CHIKV structural and non-structural proteins (Appendix A). The T-cell epitopes predicted in this study overlapped either fully or partially with those identified in the positive experimental data (Table 2, Table 3 and Appendix A). Moreover, in the E2EP3 epitope reported to be associated with the early neutralizing response to CHIKV in a study by Kam YW et al. [30], there were two major variants (^1^STKDNFNVYKATRPYLAH^18^ and ^1^S**I**KD**H**FNVYKATRPYLAH^18^) and five minor variants, which were detected at the discotope (1–13), MHC I and MHC II hot spot 1 (Figure 3B and Appendix A). The major E2EP3 sequence (S**I**KD**H**FNVYKATRPYLAH) of the Asian strain (no. ACY66830) differed from that of the ECSA strain (no. NP 690589) and IOI strain (no. EF210157).

To further explore the molecular interaction between MHC class I molecules and epitope variations, we used GalxyPepDock to compare the peptide–MHC complex (pMHC) structure (HLA-A*02; PDB ID. 3MRB) between the epitope sequence ^259^FPLANVTCMV^268^ (found in the ESCA and IOI strains) and the variant epitope sequence M267R (present in the Asian strain) (Figure 4). The best clustering model for the interaction between HLA-A*02 and ^259^FPLANVTCMV^268^ demonstrated an average RMSD of 1.12872, whereas the model for M267R exhibited an average RMSD of 2.86341 (Appendix A). Additionally, the contact map revealed distinct receptor/peptide residue interactions between the HLA-A*02-variant docking models (Figure 3 and Appendix A). Furthermore, the pMHC structures (HLA-A*30) between the two major variants (^1^STKDNFNVYK^10^ and ^1^S**I**KD**H**FNVYK^10^) were analyzed (Appendix A). The best clustering model for the interaction between HLA-A*30 and ^1^STKDNFNVYK^10^ demonstrated an average RMSD of 6.48384, whereas the model for ^1^S**I**KD**H**FNVYK^10^ exhibited an average RMSD of 0.346376 (Appendix A). Differences in receptor/peptide residue interactions are believed to contribute to variations in binding affinity for different variants.

## 4. Discussion

The E2 glycoprotein appears to be an immunodominant protein in CHIKV infection [30,31] and plays a crucial role in the initiation of viral replication and composition of the viral envelope with E1 and E3 [11,21]. There are four major CHIKV lineages (ESCA, WA, IOI, and Asian lineages) [6,9], all of which comprise one serotype with cross-reactivity [32,33]. However, whether numerous genetic mutations result in immunological variations remains unclear. In fact, differences in immunogenicity have been observed in the neutralization efficiency of convalescent human sera between ESCA and Asian strains [34].

Therefore, we endeavored to analyze the relatedness in the genetic variation and immunogenic epitopes of clonal lineages, processing a large quantum of genetic variation and immunological information (44 STs extracted from 1660 sequences). The E2 phylogenetic analysis was similar to previous epidemic studies for the geographic lineage [6,9]. This result indicates antigenic differences in the geographic lineage similar to those reported in other studies. Analyzing the phylogenetic correlation of specific STs that have emerged in particular regions, and approaching their epidemiological spread, structural, and immunological variations in an integrated manner, will be important for future research.

Based on the search for genetic variation in the E2 gene, we also investigated the B-cell and T-cell epitopes of the ECSA (clade III), IOI (clade I), and Asian (clade II) representative strains (Table 1, Table 2 and Table 3). Epitope prediction revealed conserved hot spots among clonal lineages with variation (Table 1, Table 2 and Table 3, Figure 3B).

The mAb mapping results presented in other studies identified that most of the binding domains are located in the E2 domain B (with some results also indicating domain A and the β-ribbon region) [35,36,37,38]. The B-cell linear epitopes presented in Table 1 are distributed across domains A, B, and C, with epitopes in domain B being overly broad, and epitopes were also identified in the structurally inaccessible domain C. This result appears to deviate from the findings of experimental data. The method we adopted has a limitation in that it is not based on actual experimental data. However, the goal of this study was not to discover new epitopes, but rather to provide a broad range of epitope candidates based on widely reported variants from around the world.

In T-cell epitope prediction, eight conserved MHC class I hot spots and three MHC II hot spots were predicted with the reference HLA alleles set. MHC II hot spots tend to overlap with MHC I hot spot 1 and spot 7, respectively (Figure 3B). Variation analysis of the predicted epitope showed highly frequent variations (N5H, S118G, G194S, L248F/S, and I255V/T) among clade II strains, distinct from strains of other lineages (Figure 3B).

Overlapping MHC I and II hot spot 1 includes a region known to be the dominant epitope of E2 (E2EP3) (Figure 3B). E2EP3 is located proximal to the furin E2/E3 cleavage site [39] and can elicit the neutralizing IgG3 associated with viral clearance and protection [30,33]. In this study, we identified seven variations in E2EP3, including two major variations (^1^STKDNFNVYKATRPYLAH^18^ and ^1^S**I**KD**H**FNVYKATRPYLAH^18^). Variations were found mainly at the 2nd and 5th loci such as T2I or T2V (14/44 STs) and N5H (10/44 STs). These variations were mainly found in clade II strains.

Additionally, the predicted pMHC structures of the variants exhibited different RMSD values and contact maps (Figure 4 and Appendix A). These results suggest that structural changes might serve as the molecular basis for antigen presentation with some variation and might explain the difference in neutralizing antibody efficiency among CHIKV clonal lineages.

We analyzed and compared the B- and T-cell epitope data from our study with the results of other in silico studies. Unlike our study, which focused primarily on the E2 antigen and its wide range of variants, several in silico studies targeted not only the full range of structural proteins, including E2, but also non-structural proteins of CHIKV [40,41,42]. Some studies have also focused mainly on the E1 structural protein [43], or have simultaneously analyzed both E1 and E2 structural proteins in silico studies [44].

In immunoinformatics studies on CHIKV, including this one, the research methodologies and study targets differ slightly (e.g., Mexican strain, refer to 40), leading to varying epitope results. However, some common findings were observed. For instance, the epitope ^284^IMLLYPDHPTLLSYR^298^, predicted as both a T-cell and B-cell epitope [44], overlapped with MHC I hot spot 7, MHC II hot spot 2, and the discotope region identified in our study (Figure 3B). The T-cell epitope ^94^TGTMGHFIL^102^ [40] was located in MHC I hot spot 3, and the T-cell epitopes ^1^STKDNFNVYKATRPYLAHC^19^, ^91^CTITGTMGHFILARC^105^, ^231^NHKKWQYNSPLVPRN^245^, ^276^VTYGKNQVIMLLYPD^290^, and ^326^LEVTWGNNEPYKYWP^340^ [42] overlapped with the MHC hot spots (Figure 3B) identified in our study. Differences were also observed, likely due to the use of different analytical tools and criteria, which highlight the current limitations of in silico analysis.

The CHIKV vaccine (VLA-1553) recently approved by the US FDA [15] is a live-attenuated vaccine developed based on the ESCA lineage strain. In our study, strains from clade II (including the Asian strain), which are widely distributed, showed not only variations in B-cell epitopes (Table 1) but also the most significant variations in the T-cell hot spot (Figure 3B). As previously mentioned, these variations could be closely related to differences in immunogenicity [34]. This study provides preliminary data that can support the development of vaccines targeting variant strains, or epitope-based peptide vaccines with a range of variations. Additionally, these findings may contribute to the development of diagnostic methods for antigenic variants or specific peptide pools for T-cell immunity-based diagnostics. Furthermore, this study may be relevant for the development of immune-based therapies.

This in silico approach involved processing large amounts of global data and genetic variation according to clonal lineage, followed by prediction of immunogenic hot spots with variation and analysis of pMHC structure. These data have some limitations that should be confirmed through experiments. However, conventional experimental methods for selecting vaccine candidates, immunotherapy targets, and other biomarkers have drawbacks, including selection bias, time-consuming procedures, and limitations in processing extensive genetic and immunogenic variations. Prior to the experimental process, an in silico approach can be useful for selecting biomarkers for diverse vaccines, immunotherapies, and other studies. Furthermore, predicting and evaluating immune interactions by variation of dominant epitope could be worthwhile in studying vaccination efficacy and diverse vaccine platforms such as subunit, peptide, and nucleic acid-based vaccines.

In silico predictions pose risks related to algorithm bias and virtual interpretations. Therefore, we conducted an analysis comparing actual results from the experimental database (IEDB DB) with the predicted epitope data. Most of the predicted epitopes and their amino acid positions closely or partially aligned with the actual epitopes (Table 2, Table 3, Appendix A). We believe that this approach helps to reduce the risks associated with predictive studies. Furthermore, the CHIKV E2 epitopes analyzed in the experimental database contained very limited data on antigenic variants. Our findings contribute to identifying and presenting the antigenic variants for each epitope across the clonal lineage.

Taken together, this in silico approach can serve as a complementary tool to experimental platforms for exploring immunogenic variations and developing biomarkers for vaccine design, immunotherapy, and diagnostics.

## Figures and Tables

**Figure 1 viruses-16-01689-f001:**
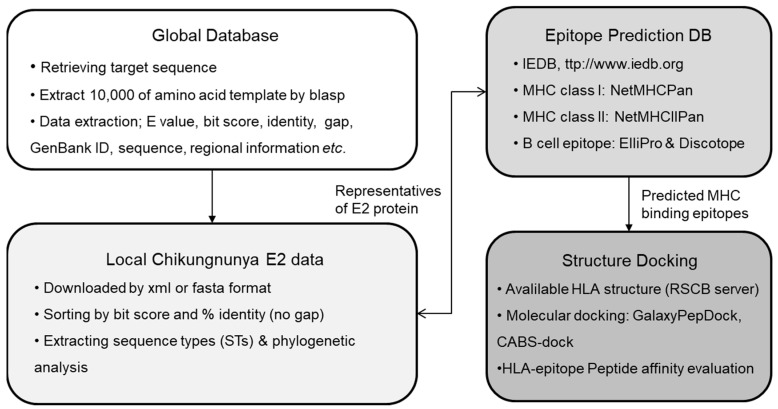
Overview of the in silico approach. Immunogenic epitopes of each E2 were analyzed by the immune epitope database and analysis resource (IEDB, http://www.iedb.org). Affinity between each HLA molecule and epitope were simulated at Galaxypepdock (https://galaxy.seoklab.org/cgi-bin/submit.cgi?type=PEPDOCK) and CABS-dock (https://biocomp.chem.uw.edu.pl/CABSdock/).

**Figure 2 viruses-16-01689-f002:**
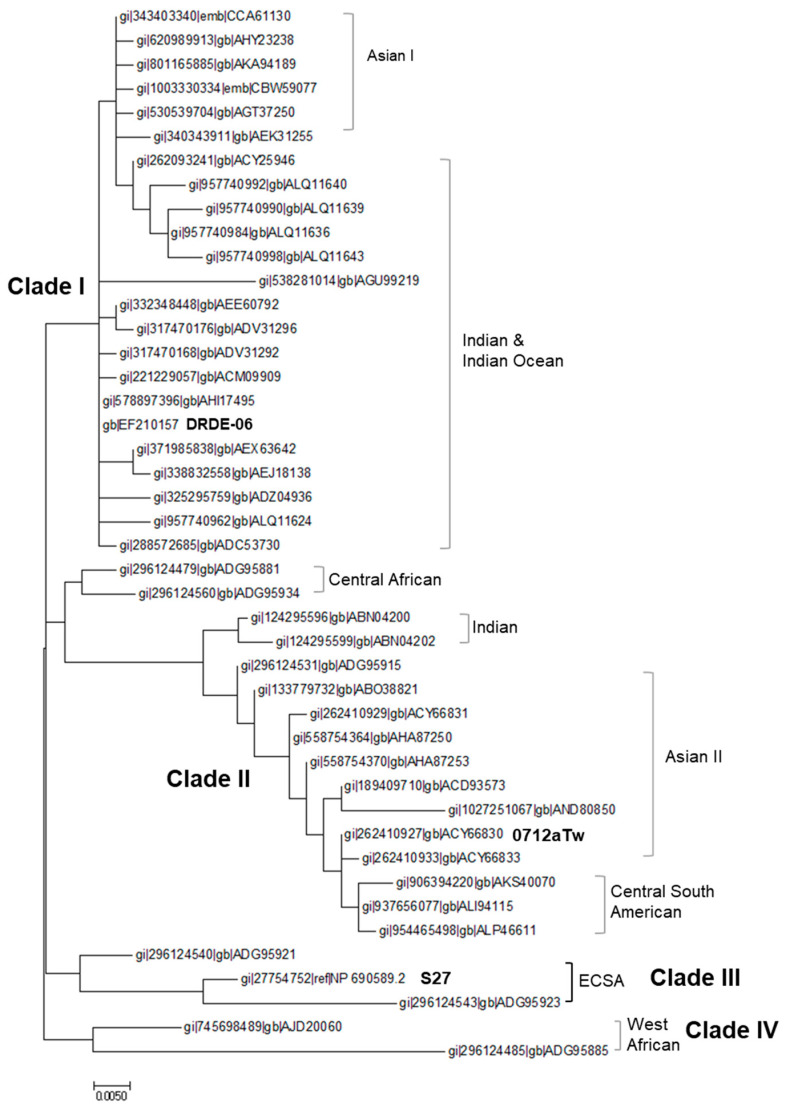
Molecular phylogenetic analysis of Chikungunya virus envelope 2 (CHIKV E2) gene (amino acids sequence) by maximum likelihood method. It shows the genetic diversity of CHIKV E2 genes and the geographic distribution of CHIKV. ECSA indicates the East Central South African strain of CHIKV. The clades were distinguished based on a 0.02 sequence difference. Three E2 candidates are indicated in the figure (DRDE-06, 0712aTw, and S27).

**Figure 3 viruses-16-01689-f003:**
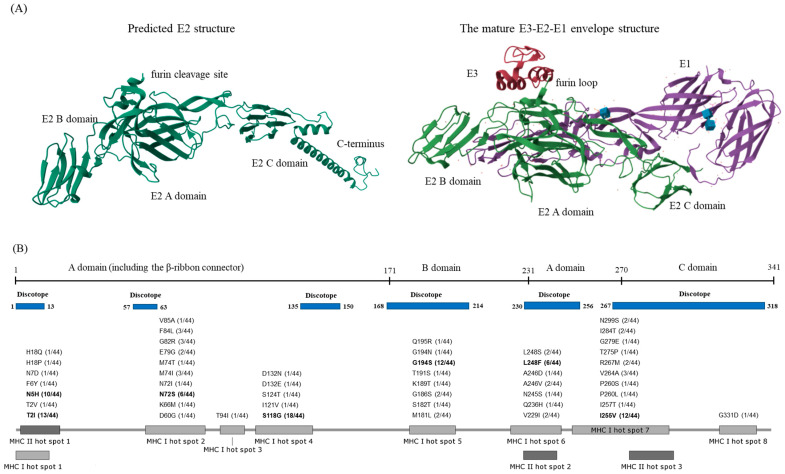
Comparison of mature envelope structure and the predicted Chikungunya virus envelope 2 (E2) structure (**A**) and the map of MHC Class I and II binding epitope hot spots predicted by NetMHCPan and NetMHCIIPan 4.0 (**B**). (**A**) The predicted structure (left) of ESCA E2 (NP 690589) is compared to the mature envelope complex (right) (PDB ID. 3N42, https://doi.org/10.2210/pdb3N42/pdb). These structures were visualized by Mol* Viewer (https://molstar.org/viewer/) [29]. The blue boxes indicate N-glycans (N-acetylglucosamine and 2-Acetamido-2-deoxy-D-glucose) meaning the glycoprotein (E1 and E2). (**B**) This map shows HLA allele binding hot spots at mature E2 amino acids. The regions corresponding to the domains A, B, and C of the mature E2 structure are displayed at the top. Additionally, blue boxes with amino acid residue numbers indicate regions with highly distributed discontinuous epitopes (discotopes). Each mutation is indicated above each hot spot (mutation frequency among 44 STs is indicated in parentheses). Bold letters indicate highly frequent mutations in E2 STs.

**Figure 4 viruses-16-01689-f004:**
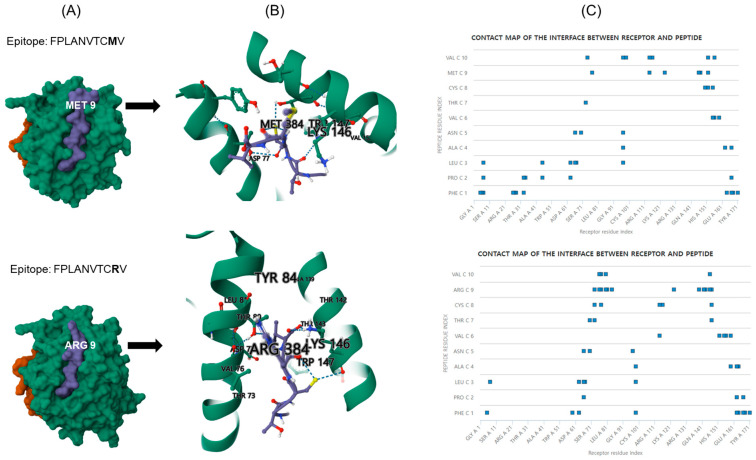
Molecular docking of E2 ^259^FPLANVTCMV^268^ of the epitope (ESCA and IOI strains) and ^259^FPLANVTCRV^268^ of the epitope (Asian strain) presentation on HLA-A2. (**A**) Simulated structures of pMHC were presented for E2 epitopes. GalaxyPepDock was used for molecular docking between pMHC and epitopes. The models were ranked by TM-score combined similarity score. Docking was presented by the molecular surface model created from Mol*Viewer, and violet surface and stick indicate epitope. (**B**) Comparison of pMHC structure between epitopes variant. The predicted interaction between pMHC and epitopes is enlarged in the variation site, which shows the interactions with different receptor residues. Contacted amino acid residues are presented in the figures. (**C**) Contact map of the interface between receptor (A polymer) and peptide (C polymer). The contact map was predicted using CABS-dock online (https://biocomp.chem.uw.edu.pl/CABSdock/).

**Table 1 viruses-16-01689-t001:** Profiling of the major predicted linear peptides for B-cell epitopes in mature E2 protein.

E2 Domain	Clade III: ECSA Strain (no. NP 690589) ^a^	Clade I: IOI Strain (no. EF210157) ^a^	Clade II: Asian Strain (no. ACY66830) ^a^
A (including β-ribbon region)	STKDNFNVYKA (1–11) ^b^	STKDNFNVYKA (1–11)	S**I**KD**H**FNVYKA (1–11)
	CGEGHSC (22–28)	CGEGHSCH (22–29)	CGEGHSC (22–28)
	TDDSHDWTK (58–66)	IKTDDSHDWTK (56–66)	IKTDDSHDWTK (56–66)
	TDSRKIS (116–122)	TDSRKIS (116–122)	TD**G**RKIS (116–122)
	RPQHGKE (144–150)	RPQHGKE (144–150)	RPQHG**R**E (144–150)
	QSNAATAEEIE (158–168)	QSTAATTEEIE (158–168)	QSTAAT**A**EEIE (158–168)
	VPRNAELGDRKGKIHI (242–257)	VPRNAELGDR**K**GKIHI (242–257)	VPRN**A**E**F**GDRKGK**V**HI (242–257)
B	MPPDTPDRTLLSQQSGNVKITVN (171–193)	MPPDTPDRTLMSQQSGNVKITVN (171–193)	MPPDTPDRTLMSQQSGNVKITVN (171–193)
	QTVRYKCNCGGSNEGLITTDKVINNCKVDQCHAAVTNHKKW (195–235)	QTVRYKCNCGGSNEGLTTTDKVINNCKVDQCHAAVTNHKKW (195–235)	QTVRYKCNCG**D**S**S**EGL**T**TTDKVINNCKVDQCHAAVTNHKKW (195–235)
C	PKARNPTVTYGK (269–280)	PKARNPTVTYGK (269–280)	PKARNPTVTYGK (269–280)

^a^: GenBank accession number, ^b^: E2 amino acid sequence number. The underlined letters indicate amino acid residues of discotopes. The bold letters indicate amino acids showing more than 10% variation within the same lineage strains.

**Table 2 viruses-16-01689-t002:** Prediction of MHC class II binding peptide in mature E2 of CHIKV representative strains by NetMHCIIPan 4.0 in IEDB.

Amino Acid Position ^a^	Conserved Core Peptide ^b^	MHC Class II HLA Allele	Shared IEDB Epitope ID ^c^	Response Measured ^d^
3–20 (E2 domain A)	YKATRPYLA	HLA-DRB1*01:01 HLA-DRB1*07:01 HLA-DRB1*09:01 HLA-DQA1*04:01/DQB1*04:02	169782, 2191694, 2252980, 2253056	IFN-gamma release Biological activation
232–246	W**Q**YNSPLVP	HLA-DRB3*02:02	2190289	IFN-gamma release
280–299 (E2 domain C)	**I**MLLYPDHP MLLYPDHPT LYPDHPTLL	HLA-DRB1*03:01 HLA-DRB1*15:01 HLA-DRB3*01:01 HLA-DRB4*01:01	2192092, 2191085, 2253044, 2190783, 2190320	IFN-gamma release Biological activation

These data were extracted by filtering at a rank below 1 percentile. Bold letters indicate amino acids with variations found among the CHIKV E2 STs (Figure 3). ^a^: E2 amino acid sequence number; ^b^: Allele binding core sequence conserved in the ESCA, IOI, and Asian strains; ^c^: IEDB epitope ID corresponds to amino acid position of predicted epitope in this study; ^d^: These data obtained from the positive experimental data of IEDB (Appendix A).

**Table 3 viruses-16-01689-t003:** Prediction of HLA-A*02 binding peptide in mature E2 of CHIKV representative strains by NetMHCPan 4.0 in IEDB.

Amino Acid Position ^a^	ECSA Strain (no. NP 690589)	Indian Strain (no. EF210157)	Asian Strain (no. ACY66830)	MHC Class II HLA Allele	Shared IEDB Epitope ID ^b^
48–56	IQVSLQ**IG**I		IQVSLQ**IG**I	HLA-A*02:06	2189548
69–78	YMD**N**H**I**PADA			HLA-A*02:03	2190586
95–103	GTMGHFILA	GTMGHFILA	GTMGHFILA	HLA-A*02:06	2189667, 2191815
180–188	L**L**SQQS**G**NV		L**M**SQQS**G**NV	HLA-A*02:03	2190966
256–269	H**I**PF**P**LAN**V** F**P**LAN**V**TC**M**V	F**P**LAN**V**TC**R**V	H**I**PF**P**LAN**V** F**P**LAN**V**TC**R**V	HLA-A*02:03	2252994 2190673
285–295	MLLYPDHPTL LLYPDHPTLL LLYPDHPTL	LLYPDHPTLL LLYPDHPTL	MLLYPDHPTL LLYPDHPTLL LLYPDHPTL	HLA-A*02:01 HLA-A*02:03 HLA-A*02:06	2191085 2253044

Data were extracted by filtering at a rank below 1 percentile. Bold letters indicate amino acids with variations found among the CHIKV E2 STs (Figure 3). ^a^: E2 amino acid sequence number, ^b^: IEDB epitope ID corresponds to amino acid position of predicted epitope in this study.

## Data Availability

The data used to support the findings of this study are available from the corresponding author upon request.

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
