# Peer review of "Predicting Immunogenic Epitopes Variation of Envelope 2 Gene Among Chikungunya Virus Clonal Lineages by an In Silico Approach"

_viruses, 2024, doi:10.3390/v16111689_

Round 1

Reviewer 1 Report

Comments and Suggestions for Authors

1.- There are no results related to section 2.2. It seems like the text in lines 138-144 is related to section 2.2. However, the experimental description of this section is extremely poor. Please explain in the methods and in the result section how these experiments were performed, the number of replicas, and the statistics used. The way this is described gives little to no information to the reader. Also, what is the point of determining that mAbs do bind to your recombinant protein? The data in Figure S1 does not provide the reader with any insight into the goal of the article.
2.- Why did you use trRosetta server over AlphaFold?
3.- Lines 155-555. Having three isolated structures does not help at all in determining if the site if the conformation of the epitopes is different among the three lineages. Please overlap the structures and determine if these epitopes are in the same position relative to each other. Also, you should look at the position of the epitopes not in the monomer structure but in the context of the trimers of heterodimers that are already published, as some of your epitopes are buried within the timer (I looked into the position of your epitopes in these structures and that is why I know some are completely buried).
4.- What criteria were used to identify the three E2 candidates? Did you generate the most common sequence of each lineage? Please be more specific.
5.- What is the role of epitopes in the C region? This region has steric clashes with E1?
6.- Are the epitopes of interest solvent accessible?  Please locate the epitopes of interest within the E2 structure in the mature virion and determine if they are solvent-accessible.
This is very important as a preliminary search showed me that most of these epitopes (Tables 1, 2, and 3) are buried inside E2 or even buried by E1. Their position could suggest that they might not be strong contributors to the immunogenicity of the protein.
If the authors think that buried epitopes still contribute please explain the rationale and any data supporting this idea. For example, the amino acids of interest in Figure 4 are mostly solvent non-accessible based on the cryo-em data in the E1/E2 dimer, hence it is not clear to me if the docking data there is relevant at all.
7.- Figure 3A is confusing. What are the blue boxes? Why are they on E1 if the subject of study is E2?
8.- It would be important to know the degree of conservation of the epitopes in Table 1, not only within its strain but also within all strains.
9.- Lines 205-210. How do you know that those epitopes are related to early neutralizing? Is this something you predict, or is it based on experimental data? Also, what do you mean by early neutralization?
10.- There are a few structures of CHIKV bound to mAbs: (2013) Elife 2: e00435-e00435, (2020) Proc Natl Acad Sci U S A 117: 27637-27645, (2023) Sci Transl Med 15: eade8273-eade8273, (2020) Proc Natl Acad Sci U S A 117: 27637-27645, (2015) Cell 163: 1095). Please discuss how your results compare to those.  More importantly, are the predicted epitopes related to those found in those experiments?
11.- What is the novelty of the results in this manuscript compared to those of: Hoque, H. et al. 2021 Helyion 7:e06396; Bappy, S.S. et al. 2020 Journal of Biomolecular Structure and Dynamics, 4:39; Islam, R. et al.  2012 Future Virology 7:10; Saeed, A., et al. 2020 Current Pharmaceutical Biotechnology 21:4 325-340; Kori, P., et al 2015 IJournal of Pharmaceutical Investigation 45, 579–591; Sanchez-Burgos, G.G. et al. 2021 Viruses 13(12):2360. Furthermore, how many of your epitopes are unique compared to those reported in these articles? If there are novel epitopes, please explain why where you were able to detect them.

I am truly intrigued by the fact that the authors do not make any reference to other articles that perform in-silico studies of CHIKV epitopes.  Without a proper discussion of how their data is novel compared to the articles I mentioned, this manuscript lacks any degree of novelty. Furthermore, with the cryo-em data that is available as well as with the AlphaFold server, they should have looked at their epitopes to know if they are in the outer surface for the trimer; it is likely that most of their sequences might not be properly presented.

Author Response

Comment 1: There are no results related to section 2.2. It seems like the text in lines 138-144 is related to section 2.2. However, the experimental description of this section is extremely poor. Please explain in the methods and in the result section how these experiments were performed, the number of replicas, and the statistics used. The way this is described gives little to no information to the reader. Also, what is the point of determining that mAbs do bind to your recombinant protein? The data in Figure S1 does not provide the reader with any insight into the goal of the article.

Response 1: We aimed to describe the screening results of E2 recombinant proteins produced from representative sequences of the CHIKV virus with the response of antibodies (Western blot). As it was a simple screening result, detailed experiments and statistical analyses were not conducted. We acknowledge that, as you mentioned, we did not provide precise information to the readers, and the data does not offer any insightful results for this paper. Therefore, we have decided to remove the related methods and results from the revised version of the manuscript.

Comment 2: Why did you use trRosetta server over AlphaFold?

Response 2: In this study, we conducted our analysis using a pipeline that integrates various tools. However, the PDB files generated by AlphaFold had limitations when used with other tools. Consequently, we utilized the trRosetta server to predict the 3D structures, and the resulting PDB files were then used for further analysis in our pipeline.

Comment 3: Lines 155-555. Having three isolated structures does not help at all in determining if the site if the conformation of the epitopes is different among the three lineages. Please overlap the structures and determine if these epitopes are in the same position relative to each other. Also, you should look at the position of the epitopes not in the monomer structure but in the context of the trimers of heterodimers that are already published, as some of your epitopes are buried within the timer (I looked into the position of your epitopes in these structures and that is why I know some are completely buried).

Response 3: Thank you for your valuable comments.

As you pointed out, B cell epitopes can vary significantly depending on structural accessibility, and thus, B cell epitope prediction can indeed be greatly influenced by the structure of the E1-E2 heterodimer.

In this study, we aimed to analyze both B cell and T cell epitopes. Therefore, we focused on the structure of the E2 antigen monomer. The representative variants of the E2 antigen in our study comprised a total of 44 sequence types (STs). Given the complexity and the significantly increased number of E1/E2 combinations, we determined that it would be challenging to include these in the current study. Additionally, as T cell epitopes are primarily evaluated based on MHC molecule binding activity, we concluded that predicting T cell epitopes using the E2 monomer alone would be sufficient without the need for the E1/E2 dimer.

However, as you rightly highlighted, structural accessibility is indeed a critical factor for B cell epitopes. In response, we reanalyzed the B cell epitopes based on the antigen structure using the Ellipro method. As a result of the reanalysis, the predicted B cell epitopes are in similar positions relative to each other across the three lineages with the variations. Accordingly, we have updated Table 1 and added the results of the B-cell discontinuous epitope prediction (Discotope) for the representative STs (ECSA, IOI, and Asian strains), specifically focusing on the overlap of Discotope results across the three STs (Figure S2).

Comment  4: What criteria were used to identify the three E2 candidates? Did you generate the most common sequence of each lineage? Please be more specific.

Response 4:  Thank you for your comment.

We have revised the text and Fiugure 2 to be more specific, as follows: The E2 candidates were selected from the four clades distinguished by a 0.02 sequence difference. We chose the E2 genotypes from clades I-III, which had the highest frequency and included the reference strain. We have updated Figure 2 and reflected this information in the main text accordingly (Lines 143-151).

Comment  5: What is the role of epitopes in the C region? This region has steric clashes with E1?

E1 and E2 of the CHIKV are known as the main antigenic determinants and form an icosahedral structure at the virion surface.

Response 5:  As you commented, the cryo-em data in the E1/E2 dimer showed that domain C of E2 is sandwiched between domain II of two E1 molecules, making the walls of a central cavity under domain A.

We predicted and pooled potential B and T cell epitopes of E2 using in silico methods. Although this study had limitations in precisely determining the role of domain C epitopes, we recognize its significance. Domain C is a critical region that forms a structural interface with E1 and contributes to cavity formation, suggesting it has potential as a T cell epitope candidate.

Comment  6: Are the epitopes of interest solvent accessible? Please locate the epitopes of interest within the E2 structure in the mature virion and determine if they are solvent-accessible.

This is very important as a preliminary search showed me that most of these epitopes (Tables 1, 2, and 3) are buried inside E2 or even buried by E1. Their position could suggest that they might not be strong contributors to the immunogenicity of the protein.

If the authors think that buried epitopes still contribute please explain the rationale and any data supporting this idea. For example, the amino acids of interest in Figure 4 are mostly solvent non-accessible based on the cryo-em data in the E1/E2 dimer, hence it is not clear to me if the docking data there is relevant at all.

Response 6: As you mentioned, assessing the solvent accessibility of epitopes is a crucial factor in predicting B cell epitopes. In response to your comments, we reanalyzed the B cell epitopes, taking structural accessibility into account (specifically focusing on discontinuous epitope analysis). Accordingly, we revised Table 1 and added Figure S2, with corresponding updates to the manuscript.

In this analysis, we also predicted B cell epitopes based on the monomer structure, which allowed us to identify epitopes in domains A, B, and C. The selection of epitopes in the relatively buried C domain (E1) of the E1/E2 dimer structure was one of the limitations of this study, which we have addressed in the Discussion section.

However, in predicting T cell epitopes, where the ability to bind to MHC molecules is the key factor, we believe that the heterodimer structure is not a significant concern.

Regarding the molecular docking data, since it mainly discusses changes in binding affinity due to variations in T cell epitopes, we cautiously concluded that structural non-accessibility may not pose a major issue in this context.

Comment 7: Figure 3A is confusing. What are the blue boxes? Why are they on E1 if the subject of study is E2?

Response 7: In Figure 3A, we aimed to demonstrate that the E2 structure model predicted in this study shows similarity to the E2 structure within the mature envelope complex (E3-E2-E1, analyzed by X-ray diffraction). We used the generated 3D structure file to predict both B cell and T cell epitopes.

The blue boxes indicate N-glycans (N-acetylglucosamine and 2-Acetamido-2-deoxy-D-glucose) meaning the glycoprotein (E1 & E2). It was described in the legend of Figure 3A in the revised manuscripts.

Comment  8: It would be important to know the degree of conservation of the epitopes in Table 1, not only within its strain but also within all strains.

Response 8: In Table 1, which includes the B cell epitope candidates from the reanalysis, most epitopes were conserved, with the exception of a few variations. Amino acids showing more than 10% variation within the same lineage strains have been highlighted.

Comment  9: Lines 205-210. How do you know that those epitopes are related to early neutralizing? Is this something you predict, or is it based on experimental data? Also, what do you mean by early neutralization?

Response 9: Kam YW et al. analyzed epitopes as serological markers by screening plasma samples obtained from CHIKV-infected patients during the early convalescent phase. They reported that anti-E2EP3-specific antibodies exhibit neutralizing effects and provide protection in preclinical studies (reference: Kam YW et al., Early neutralizing IgG response to Chikungunya virus in infected patients targets a dominant linear epitope on the E2 glycoprotein. EMBO Mol. Med. 2012, 4, 330-343). The term "early neutralization" is used in this context and is discussed in the manuscript.

Comment 10: There are a few structures of CHIKV bound to mAbs: (2013) Elife 2: e00435-e00435, (2020) Proc Natl Acad Sci U S A 117: 27637-27645, (2023) Sci Transl Med 15: eade8273-eade8273, (2020) Proc Natl Acad Sci U S A 117: 27637-27645, (2015) Cell 163: 1095). Please discuss how your results compare to those.  More importantly, are the predicted epitopes related to those found in those experiments?

Response 10: Thank you very much for pointing out the aspects we had not fully considered. Your suggested references allowed us to gain a deeper understanding of the E2-E1 heterodimer structure involved in mAbs binding.

The mAb mapping presented in your referenced paper identified that most of the binding domains are located in the E2 domain B (with some results also indicating domain A and the β-ribbon region). Nearly all of these were included in the revised Table 1.

Our study aims to identify global strain variants and propose B cell epitope candidates. We acknowledge that the method we adopted has the limitation of not being based on actual experimental data. However, the goal of this study was not to discover new epitopes, but rather to provide a broad range of epitope candidates based on widely reported variants from around the world.

We have now included these limitations and comparisons to experimental data in the Discussion section (lines 283-291).

Comment  11: What is the novelty of the results in this manuscript compared to those of: Hoque, H. et al. 2021 Helyion 7:e06396; Bappy, S.S. et al. 2020 Journal of Biomolecular Structure and Dynamics, 4:39; Islam, R. et al.  2012 Future Virology 7:10; Saeed, A., et al. 2020 Current Pharmaceutical Biotechnology 21:4 325-340; Kori, P., et al 2015 IJournal of Pharmaceutical Investigation 45, 579–591; Sanchez-Burgos, G.G. et al. 2021 Viruses 13(12):2360. Furthermore, how many of your epitopes are unique compared to those reported in these articles? If there are novel epitopes, please explain why where you were able to detect them.

Response 11: As you mentioned, we compared and analyzed the B cell and T cell epitope data from our study with the results presented in the referenced papers. Unlike our study, which focused primarily on the E2 antigen and its wide range of variants, several in silico studies targeted not only the full range of structural proteins, including E2, but also the non-structural proteins of CHIKV (Hoque, H. et al.; Kori, P., et al.; Sanchez-Burgos, G.G. et al.). There are also studies that focused mainly on the E1 structural protein (Saeed, A., et al.) or simultaneously analyzed both E1 and E2 structural proteins in in silico studies (Bappy, S.S. et al.).

In immune-informatics studies on CHIKV, including this one, the research methodologies and study targets differ slightly (e.g., Mexican strain in Sanchez-Burgos, G.G. et al.), leading to varying epitope results. However, some common findings can be drawn. For instance, the epitope 284IMLLYPDHPTLLSYR298, predicted as both a T cell and B cell epitope (Bappy, S.S. et al.), overlaps with the MHC I hotspot 7, MHC II hotspot 2, and discotope region identified in our study (Figure 3B). The T cell epitope 94TGTMGHFIL102 (Kori, P., et al.) was found to be located in our MHC I hotspot 3, and the T cell epitopes 1STKDNFNVYKATRPYLAHC19, 91CTITGTMGHFILARC105, 231NHKKWQYNSPLVPRN245, 276VTYGKNQVIMLLYPD290, and 326LEVTWGNNEPYKYWP340 (Sanchez-Burgos, G.G. et al.) overlapped with MHC hotspots (Figure 3B) identified in our study. There are also differences observed, likely due to the use of different analysis tools and criteria, which highlight the current limitations of in silico analysis.

We revised the Discussion section lines 310-326.

Reviewer 2 Report

Comments and Suggestions for Authors

With regard to your submission of the article entitled "Prediction of Variants in the Immunogenic Epitope of the E2 Gene Between Chikungunya Virus Clonal Lines by In Silico Simulations", I have read the manuscript and materials provided, and based on this information with my expertise, I would like to raise following comments:

This manuscript provides an interesting study on the immunogenic variability of the Chikungunya virus (CHIKV) envelope 2 (E2) gene. The work aims to explore the genetic diversity of this crucial component in the anti-CHIKV immune response using an in silico approach.

Weaknesses/Suggestions for Improvement:

1# The abstract lacks clarity in explaining the specific methodology and results achieved. Clarifying the key findings and their implications would strengthen the abstract.

2#More details on the in silico approach and its limitations should be provided to allow readers to fully assess the study's rigor and reproduction. Such as further clarifying the detailed criteria for data screening and processing to ensure the reliability and accuracy of the analysis results.

3# The discussion section could be expanded to compare the findings with previous studies and discuss potential implications for vaccine development or diagnostic tools.In addition, since the further research directions of the results predicted and screened in this study are predetermined in this section, whether the preliminary experimental verification results should be appropriately added in this paper to enhance the reliability and persuasiveness of the results of this study.

4# The manuscript would benefit from providing additional information on the clinical significance of the identified immunogenic epitopes and their potential use in immune-based therapies.

5# With some clarifications and expansions to the methodology, results, and discussion sections, the manuscript could be suitable for major revision, more experimental verification results should be added. The authors should also carefully consider the clinical implications of their findings and discuss their potential applications.

Comments on the Quality of English Language

Minor editing of English language required

Author Response

Comment 1: The abstract lacks clarity in explaining the specific methodology and results achieved. Clarifying the key findings and their implications would strengthen the abstract.

 Response 1: Thank you very much for your comments. As you suggested, we have revised and strengthened the Abstract sections accordingly (Abstract section, line 21-24 & line 2-29).

Comment 2: More details on the in silico approach and its limitations should be provided to allow readers to fully assess the study's rigor and reproduction. Such as further clarifying the detailed criteria for data screening and processing to ensure the reliability and accuracy of the analysis results.

Response 2: We have revised the Methods section to provide further details regarding the data screening and processing in the in silico approach (Lines 91-107 and 114-119). To enhance the data, we have also included a detailed description of the phylogenetic analysis results for the representative genotypes (Figure 2 and Lines 143-151). Additionally, in this revised version, we reanalyzed the B cell epitopes using a structure-based method (Table 1 and Lines 171-175). We have also added Figure S2 related to B cell discontinuous epitopes and revised Figure 3 accordingly.

Comment 3: The discussion section could be expanded to compare the findings with previous studies and discuss potential implications for vaccine development or diagnostic tools. In addition, since the further research directions of the results predicted and screened in this study are predetermined in this section, whether the preliminary experimental verification results should be appropriately added in this paper to enhance the reliability and persuasiveness of the results of this study.

Response 3: Thank you for your valuable insights. We have incorporated the information and discussion regarding previous structure-based mAbs mapping studies and relevant in silico research into the Discussion section (Lines 282-291 and 310-326).

Comment 4: The manuscript would benefit from providing additional information on the clinical significance of the identified immunogenic epitopes and their potential use in immune-based therapies.

Response 4: We have added revisions to the Discussion section addressing the issue of major variants with different immune epitopes compared to the currently approved CHIKV vaccine, as well as our study's findings (lines 327-337).

Comment 5: With some clarifications and expansions to the methodology, results, and discussion sections, the manuscript could be suitable for major revision, more experimental verification results should be added. The authors should also carefully consider the clinical implications of their findings and discuss their potential applications.

Response 5: Thank you for your valuable advice. In this revised version, we have made additional revisions to the Methods, Results, and Discussion sections. The revised sections are highlighted in red in the manuscript.

Reviewer 3 Report

Comments and Suggestions for Authors

The paper titled "Predicting Immunogenic Epitopes Variation of Envelope 2 Gene among Chikungunya Virus Clonal Lineages by an In Silico Approach" is a study that employs computational methods to investigate the immunogenic variability of the Chikungunya virus (CHIKV) envelope 2 (E2) gene. The research seeks to elucidate the impact of genetic variations across different clonal lineages of CHIKV on immunogenic epitopes, a factor of significant importance for the development of vaccines and diagnostic tools. Nonetheless, the manuscript necessitates additional refinements to enhance its quality and clarity.

1. It would be beneficial to consider additional empirical studies to enhance the validation of the findings' robustness.

2. It may be advantageous to provide additional insights following molecular docking to better articulate the results and explore their biological implications.

3. The manuscript could be further strengthened by including specific descriptive details, such as the source of the E2 gene sequences and the rationale for the selection of the three E2 candidates.

4. It would be valuable to expand the phylogenetic analysis to identify distinct sequences influenced by regional viral characteristics and to perform subsequent analyses to understand their interplay in structural and functional aspects.

Author Response

Comment 1: It would be beneficial to consider additional empirical studies to enhance the validation of the findings' robustness.

Response 1: Thank you for your valuable comments. In the revised Discussion section, we have incorporated a comparative analysis of our findings with previous mAb mapping studies based on cryo-EM data and other related studies (Lines 282-291 and 310-326). Additionally, as a preliminary study, we have further discussed the limitations of our research and potential future applications (lines 327-337).

Comment 2: It may be advantageous to provide additional insights following molecular docking to better articulate the results and explore their biological implications.

We have added the following to the Results section: The differences in receptor/peptide residue interactions are believed to contribute to variations in binding affinity with different variants. Additionally, the binding affinity differences between HLA molecules and the variants were also analyzed (lines 247-249).

Comment 3: The manuscript could be further strengthened by including specific descriptive details, such as the source of the E2 gene sequences and the rationale for the selection of the three E2 candidates.

Response 3: As you pointed out, we have revised Figure 2 to more specifically clarify the source of the E2 gene, and we have provided additional details in the Results section (Lines 143-151).

Comment 4: It would be valuable to expand the phylogenetic analysis to identify distinct sequences influenced by regional viral characteristics and to perform subsequent analyses to understand their interplay in structural and functional aspects.

Response 4: As you mentioned, analyzing the phylogenetic relationships of specific sequence types (STs) that have emerged in particular regions, and approaching their epidemiological spread, structural, and immunological variations in an integrated manner, will be important for future research. We have added this point to the Discussion section.

Round 2

Reviewer 1 Report

Comments and Suggestions for Authors

While I believe that the impact and originality of this article are low and contribute very little to the field, the authors did most of the modifications I suggested.

Author Response

# Reviwer 1 comments;

While I believe that the impact and originality of this article are low and contribute very little to the field, the authors did most of the modifications I suggested.

Response;

We appreciate your comments and have taken them into careful consideration.

This manuscript focuses on studying antigenic epitope variations across various lineages of strains. From a large dataset, we extracted representative sequence types and analyzed the epitope variations of these representative antigens, reporting a significant increase in epitope variation in certain clades.

It is true that experimental research on such variations can be both time-consuming and costly. Our study offers an alternative approach by processing a vast amount of global data in a relatively short period, allowing us to identify key antigenic variations that can complement experimental data.

In order to address the limitations raised by the reviewers, we compared the epitopes predicted in our study with actual experimental results (ELISA, ELISPOT, etc.) during the revision process. Most of the predicted epitopes corresponded to positive results from assays registered in IEDB. The CHIKV E2 epitopes analysed in the experimental database had very limited data on antigenic variants. We believe that our results contribute to presenting the antigenic variants for each epitope across the clonal lineage. We have updated the results and tables (Tables 2 and 3) in the manuscript and provided additional data in the supplementary tables (Tables S1, S2, S6).

Additionally, to provide more detailed genetic information for each sequence type (ST) to readers, we included a supplementary figure (Figure S1) displaying mature E2 variations in STs.

We hope that our analysis of epitope variations will contribute to future research on vaccine development and diagnostic antigens in response to antigenic variations.

Reviewer 2 Report

Comments and Suggestions for Authors

The comments in the last round has been almost addressed. 

But there are still many issues to be concerned.

the Methods and Results sections lack of wet-lab corresponding contents. What happened to the SDS-PAGE and western blot?

The most of conclusions are based on predictions. So the reliability of this work is on risk of algorithm bias and virtual interpretations. More solicited validations are urgently needed.

Comments on the Quality of English Language

Minor editing of English language required.

Author Response

Comment 1: the Methods and Results sections lack of wet-lab corresponding contents. What happened to the SDS-PAGE and western blot?

Response;

We sincerely appreciate your valuable comments.

Based on feedback from one of the reviewers, we removed the SDS-PAGE and western blot results in the first revision. However, upon further consideration, we believe that presenting the differences in antigen reactivity of the mAbs would be beneficial. Therefore, we have carefully reinstated these results in the Methods and Results sections of the manuscript (lines 81-87 and 178-185), as well as in the supplementary file (Figure S2) in the 2nd revised version.

Comment 2: Most of the conclusions are based on predictions. So the reliability of this work is on risk of algorithm bias and virtual interpretations. More solicited validations are urgently needed.

Response;

Thank you for your insightful comments.

I agree with your point that in silico predictions pose risks related to algorithm bias and virtual interpretations. Therefore, during the second revision, we conducted an analysis comparing actual results from the experimental database (IEDB DB) with the predicted epitope data. The findings have been incorporated into the main text (line 210-216, line 272-276, and line 399-406), updating Tables 2 and 3, and summarised experimental data on the E2 epitope have been added to the supplementary files (Table S1: B cell linear epitopes; Table S2: B cell discontinuous epitopes; Table S3: T cell epitopes). The predicted epitopes and their amino acid positions closely matched the actual epitopes. We believe that this approach helps to mitigate the risks associated with predictive studies. Additionally, the CHIKV E2 epitopes analyzed in the experimental database had very limited data on antigenic variants. We believe that our results contribute to presenting the antigenic variants for each epitope across the clonal lineage.

In this study, our focus was not on discovering new epitopes, but rather on tracking antigenic variants across different lineages. We hope that the findings from this research will contribute to the development of vaccines and diagnostic antigens capable of covering these antigenic variants.

The above details have been included in the results and discussion sections.

Thank you once again for your valuable comments.

For this second revision, we updated the methods section to provide more detailed information to the readers (line 69-144), and we had the manuscript professionally proofread through an English language editing service (Editage, www.editage.co.kr). All corrected parts have been marked in red text.